# Query Rewriting for Retrieval-Augmented Large Language Models

**Xinbei Ma**[1,2,*] , **Yeyun Gong**[3, #, †], **Pengcheng He**[4, #], **Hai Zhao**[1,2,†], **Nan Duan**[3]

[1]Department of Computer Science and Engineering, Shanghai Jiao Tong University
[2]Key Laboratory of Shanghai Education Commission for Intelligent Interaction
and Cognitive Engineering, Shanghai Jiao Tong University
[3]Microsoft Research Asia [4]Microsoft Azure AI

`sjtumaxb@sjtu.edu.cn, zhaohai@cs.sjtu.edu.cn,`
`{yegong, nanduan}@microsoft.com, Herbert.he@gmail.com`

## Abstract

Large Language Models (LLMs) play powerful, black-box readers in the *retrieve-then-read* pipeline, making remarkable progress in knowledge-intensive tasks. This work introduces a new framework, *Rewrite-Retrieve-Read* instead of the previous *retrieve-then-read* for the retrieval-augmented LLMs from the perspective of the query rewriting. Unlike prior studies focusing on adapting either the retriever or the reader, our approach pays attention to the adaptation of the search query itself, for there is inevitably a gap between the input text and the needed knowledge in retrieval. We first prompt an LLM to generate the query, then use a web search engine to retrieve contexts. Furthermore, to better align the query to the frozen modules, we propose a trainable scheme for our pipeline. A small language model is adopted as a trainable rewriter to cater to the black-box LLM reader. The rewriter is trained using the feedback of the LLM reader by reinforcement learning. Evaluation is conducted on downstream tasks, open-domain QA and multiple-choice QA. Experiments results show consistent performance improvement, indicating that our framework is proven effective and scalable, and brings a new framework for retrieval-augmented LLM [1].

## 1 Introduction

Large Language Models (LLMs) have shown remarkable abilities for human language processing and extraordinary scalability and adaptability in few- or zero-shot settings.(Ouyang et al., 2022; Brown et al., 2020; Chowdhery et al., 2022). However, the training process depends on large-scale high-quality corpora but without the perception of the real world. Thus, LLMs still have to face the issue of hallucination (Yao et al., 2023; Bang et al., 2023) and temporal misalignment (Röttger and Pierrehumbert, 2021; Luu et al., 2022; Jang et al., 2022). This affects the reliability of LLMs and hinders wider practical application, because the consistency between the LLM responses with the real world needs further validation. Existing work has proved that incorporating external knowledge (i.e., non-parametric knowledge) with internal knowledge (i.e., parametric knowledge) can effectively alleviate hallucination, especially for knowledge-intensive tasks. In fact, retrieval-augmented LLMs have been shown so effective that they have been regarded as a standard solution to alleviate the factuality drawbacks in naive LLM generations. Retrieval augmentation is applied to select relative passages as external contexts for the language model, which is *retrieve-then-read* framework (Lewis et al., 2020b; Karpukhin et al., 2020; Izacard et al., 2022). Take the open-domain Question-Answering task (open-domain QA) as an example, a retriever first searches for related documents for a question. Then the LLM receives the question and the documents, then predicts an answer.

As most LLMs are only accessible through inference APIs, they play the part of black-box frozen readers in the pipeline. This makes previous retrieval augmentation methods that require complete access (Lewis et al., 2020b; Guu et al., 2020; Izacard et al., 2022) no longer feasible. Recent studies on retrieval-augmented language models lean more on the LLM-oriented adaptation. An idea is to train a dense retrieval model to cater to the frozen language model (Shi et al., 2023). By using feedback from the LLM as a training objective, the retrieval model is tuned for better LLM input contexts. Another research line focuses on the design of interactions between the retriever and the reader (Yao et al., 2023; Khattab et al., 2022), where both the

---

∗ Work done during an internship at [3]Microsoft Research Asia. # Equal contribution. †Corresponding author.

This paper was partially supported by Joint Research Project of Yangtze River Delta Science and Technology Innovation Community (No. 2022CSJGG1400).

[1]https://github.com/xbmxb/RAG-query-rewriting

retriever and the reader are usually frozen. The idea is to trigger the emergent ability through carefully crafted prompts or a sophisticated prompt pipeline. Multiple interactions with external knowledge allow the LLM to approach the correct answer step by step.

However, there are still problems remaining to be solved. Existing approaches overlook the adaptation of the query, i.e., the input of the *retrieve-then-read* pipeline. The retrieval query is either original from datasets or directly determined by the black-box generation, thus is always fixed. However, there is inevitably a gap between the input text and the knowledge that is really needed to query. This limits performance and places a burden on retrieval capability enhancement and prompt engineering.

In consideration of this issue, this paper proposes *Rewrite-Retrieve-Read*, a new framework for retrieval augmentation, which can be further tuned for adapting to LLMs. In front of the retriever, a step of *rewriting the input* is added, filling the gap between the given input and retrieval need, as is shown in Figure 1. We adopt the off-the-shelf tool, an internet search engine, as the retriever, which avoids the maintenance of the search index and can access up-to-date knowledge (Lazaridou et al., 2022). Different from previous studies (Khattab et al., 2022; Yao et al., 2023) that require the memory of multiple interaction rounds between the retriever and the LLM for each sample, the motivation of our rewriting step is to clarify the retrieval need from the input text.

We also propose a trainable scheme for our *rewrite-retrieve-read* framework (Figure 1 (c)). The black-box retriever and the reader form a frozen system. To further smooth the steps of our pipeline, we apply a small, trainable language model to perform the rewriting step, denoted as the *rewriter*. The rewriter is trained by reinforcement learning using the LLM performance as a reward, learning to adapt the retrieval query to improve the reader on downstream tasks.

Our proposed methods are evaluated on knowledge-intensive downstream tasks including open-domain QA (HotpoQA (Yang et al., 2018), AmbigNQ (Min et al., 2020), PopQA (Mallen et al., 2022)) and multiple choice QA (MMLU (Hendrycks et al., 2021)). The experiments are implemented on T5-large (Raffel et al., 2020) as the rewriter, ChatGPT (Ouyang et al., 2022) and

Vicuna-13B (Chiang et al., 2023) as the LLM reader. The results show that query rewriting consistently improves the retrieve-augmented LLM performance. The results also indicate that the smaller language model can be competent for query rewriting.

To sum up, our proposed novel retrieval-augmentation method, *rewrite-retrieve-read* is the first framework where the input text is adapted for the frozen retriever and LLM reader. We introduce a tuneable scheme with a small, trainable model, achieving performance gains with less resource consumption.

## 2   Related Work

### 2.1   Retrieval Augmentation

Language models require external knowledge to alleviate the factuality drawbacks. Retrieval augmentation has been regarded as the standard effective solution. With a retrieval module, related passages are provided to the language model as the context of the original input. Thus factual information like common sense or real-time news helps with output prediction through contextualized reading comprehension.

Earlier studies use sparse retriever (Chen et al., 2017) or dense retriever (Karpukhin et al., 2020) in front of a pre-trained language model (PrLM). The neural retriever and reader are both PrLMs of trainable size like BERT (Devlin et al., 2019) or BART (Lewis et al., 2020a). Hence, the whole *retrieve-then-reader* framework is a tuneable end-to-end system, where the retrieved contexts can be regarded as the intermediate results (Karpukhin et al., 2020; Lewis et al., 2020b). Approaches to smooth the two-step framework are proposed to optimize the retrieval and the reading comprehension (Sachan et al., 2021; Lee et al., 2022; Jiang et al., 2022). More recently, retrieval remains a powerful enhancement as the size of models and data scales rapidly (Mallen et al., 2022; Shi et al., 2023; Brown et al., 2020). On the other hand, retrieval enhancement can compensate for the shortfall in parameter size, compared to large-scale language models. For example, by jointly training the retriever and the reader, Atlas (Izacard et al., 2022) shows few-shot performance on par with 540B PalM (Chowdhery et al., 2022) but be of $50\times$ smaller size.

**The Internet as a knowledge base**   More related to our work, the search engine can assume the role of the retriever and use the Internet as the source of

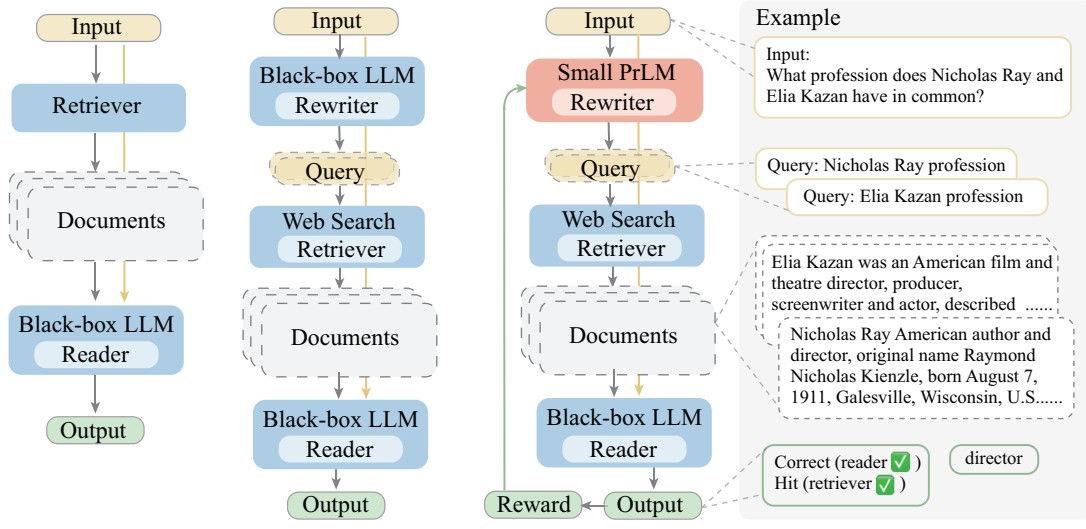

(a) Retrieve-then-read    (b)Rewrite-retrieve-read    (c) Trainable rewrite-retrieve-read

Figure 1: Overview of our proposed pipeline. From left to right, we show (a) standard *retrieve-then-read* method, (b) LLM as a query rewriter for our *rewrite-retrieve-read* pipeline, and (c) our pipeline with a trainable rewriter.

external knowledge. Komeili et al. (2022) use an internet search for relevant information based on the dialogue history to perform dialogue response generation. SeeKeR (Shuster et al., 2022) use a single Transformer to iteratively perform search query generation, then knowledge extraction for dialogue generation and sentence completion. For large-scale models, web search still shows effective for knowledge augmentation (Lazaridou et al., 2022), fact-checking (Menick et al., 2022), and LLM agent enhancement (Yao et al., 2023).

## 2.2 Cooperation with Black-box LLMs

Large Language Models, such as ChatGPT (Ouyang et al., 2022), Codex (Chen et al., 2021), PaLM (Chowdhery et al., 2022), emerge impressive natural language processing ability as well as remarkable scalability. This leads to a tendency to embrace LLMs on a wide range of NLP tasks. However, LLMs are only accessible as a black box in most cases, which is because (i) Some like Chat-GPT are not open-source and kept private; (ii) The large parameter scale requires computational resources that are not always affordable to users. This constraint means nothing is available except input and output texts.

Existing studies have proved that LLMs' abilities can be better leveraged by carefully designed interaction methods. GenRead (Yu et al., 2023) prompts an LLM to generate context instead of deploying a retriever, showing that LLMs can retrieve internal knowledge by prompting. ReAct

(Yao et al., 2023) and Self-Ask (Press et al., 2022) combines the Chain-of-Thought (CoT) (Wei et al., 2022; Wang et al., 2022) and inter-actions with web APIs. Only relying on prompt construction, Re-Act provides novel baselines for interactive tasks. Demonstrate–Search–Predict (DSP) (Khattab et al., 2022) defines a sophisticated pipeline between an LLM and a retriever. Unlike ReAct, DSP integrates prompts for demonstration bootstrap besides multi-hop breakdown and retrieval.

Despite the promising performance in the zero or few-shot setting, the behavior of LLMs sometimes needs adjustments. A feasible approach is to append trainable small models in front of or after the LLM. The small models, as a part of the parameters of the system, can be fine-tuned for optimization. RePlug (Shi et al., 2023) is proposed to fine-tune a dense retriever for the frozen LLM in the *retrieve-then-read* pipeline. The retriever is trained under the LLM's supervision to retrieve documents that are suitable for the LLM. With the same purpose, Directional Stimulus Prompting (Li et al., 2023) deploys a small model to provide the LLM with stimulus (e.g., keywords for summarization, or dialogue actions for response generation), which is updated according to the LLM reward.

Different from the inspiring work mentioned above, our proposed pipeline contains a query rewriting step in front of the *retrieve-then-read* module. We further propose a trainable scheme with a small rewriting model, which is a novel enhancement for retrieval-augmented LLM by re-

constructing the search query.

## 3 Methodology

We present *Rewrite-Retrieve-Read*, a pipeline that improves the retrieval-augmented LLM from the perspective of query rewriting. Figure 1 shows an overview. This section first introduces the pipeline framework in section 3.1, then the trainable scheme in section 3.2.

### 3.1 *Rewrite-Retrieve-Read*

A task with retrieval augmentation can be denoted as follows. Given a dataset of a knowledge-intensive task (e.g., open-domain QA), $D = \{(x, y)_i\}, i = 0, 1, 2, \ldots, N$, $x$ (e.g., a question) is the input to the pipeline, $y$ is the expected output (e.g., the correct answer). Our pipeline consists of three steps. (i) Query rewrite: generate a query $\tilde{x}$ for required knowledge based on the original input $x$. (ii) Retrieve: search for related context, *doc*. (iii) Read: comprehend the input along with contexts $[doc, x]$ and predict the output $\hat{y}$.

A straightforward but effective method is to ask an LLM to rewrite queries to search for information that is potentially needed. We use a few-shot prompt to encourage the LLM to think, and the output can be none, one or more queries to search.

### 3.2 Trainable Scheme

Besides, total reliance on a frozen LLM has shown some drawbacks. Reasoning errors or invalid search hinders the performance (Yao et al., 2023; BehnamGhader et al., 2022). On the other hand, retrieved knowledge may sometimes mislead and compromise the language model (Mallen et al., 2022). To better align to the frozen modules, it is feasible to add a trainable model and adapt it by taking the LLM reader feedback as a reward.

Based on our framework, we further propose to utilize a trainable small language model to take over the rewriting step, as is shown in the right part of Figure 1. The trainable model is initialized with the pre-trained T5-large (770M) (Raffel et al., 2020), denoted as *trainable rewriter*, $G_\theta$. The rewriter is first trained on pseudo data to warm up (§3.2.1), then continually trained by reinforcement learning (§3.2.2).

### 3.2.1 Rewriter Warm-up

The task, query rewriting, is quite different from the pre-training objective of sequence-to-sequence generative models like T5. First, we construct a pseudo dataset for the query rewriting task. Inspired by recent distillation methods (Hsieh et al., 2023; Ho et al., 2022), we prompt the LLM to rewrite the original questions $x$ in the training set and collect the generated queries $\tilde{x}$ as pseudo labels. The collected samples are then filtered: Those that get correct predictions from the LLM reader are selected into the warm-up dataset, denoted as $D_{Train} = \{(x, \tilde{x}) | \hat{y} = y\}$. The rewriter $G_\theta$ is fine-tuned on $D_{Train}$ with the standard log-likelihood as the training objective, denoted as

$$\mathcal{L}_{warm} = -\sum_t log p_\theta(\hat{\tilde{x}}_t \mid \tilde{x}_{<t}, x). \quad (1)$$

The rewriter model after warm-up shows modest performance, which depends on the pseudo data quality and rewriter capability. Highly relying on the human-written prompt line, $\tilde{x}$ can be suboptimal. The relatively small scale of the rewriter size is also a limitation of the performance after the warm-up. Then we turn to reinforcement learning to align the rewriter to the following retriever and LLM reader.

### 3.2.2 Reinforcement Learning

To further fine-tune the rewriter to cater to the LLM reader, we adopt a policy gradient reinforcement learning framework.

**Task Formulation** In the context of reinforcement learning, the rewriter optimization is formulated as a Markov Decision Process 5-tuple $\langle \mathcal{S}, \mathcal{A}, P, R, \gamma \rangle$. (i) The state space $\mathcal{S}$ is a finite set limited by the vocabulary and the sequence length. (ii) The action space $\mathcal{A}$ is equals to the vocabulary. (iii) The transition probability $P$ is determined by the policy network, which is the rewriter model $G_\theta$. (iv) The reward function $R$ gives a reward value that depends on the current state. The policy gradient is derived from rewards, used as the training objective. (v) $\gamma$ denotes the discount factor. More specifically, the rewriter $G_\theta$ after the warm-up is the initial policy model $\pi_0$. At each step $t$, the action $a_t$ is to generate the next token $\hat{\tilde{x}}_t$ based on the observation of the present state, $s_t = [x, \hat{\tilde{x}}_{<t}]$. When the generation is stopped by the End-Of-Sentence token, one episode is ended. After finishing the retrieval and reading, a reward is computed by evaluating the final output, i.e., a score for the LLM reader prediction.

**Policy Optimization** We adopt Proximal Policy Optimization (PPO) (Schulman et al., 2017), following (Ramamurthy et al., 2022). Maximization

of the expectation of the reward $R$ is formulated as

$$\max_{\theta} \mathbb{E}_{\hat{\tilde{x}} \sim p_\theta(\cdot | x)}[R(x, \hat{\tilde{x}})],$$

$$\max_{\theta} \mathbb{E}_{(s_t, a_t) \sim \pi_{\theta'}}[min\{k_{t,\theta} A^{\theta'}(s_t, a_t);$$
$$\mathrm{clip}\,(k_{t,\theta}, 1 - \varepsilon, 1 + \varepsilon)\, A^{\theta'}(s_t, a_t)\}], \quad (2)$$

$$k_{t,\theta} = \frac{p_\theta(a_t \mid s_t)}{p_{\theta'}(a_t \mid s_t)},$$

where $\theta'$ is the temporarily fixed policy for sampling and $\theta$ is updated. $A$ denotes the advantage function, which is formulated based on the estimation of value network $V_\phi$. The value network $V_\phi$ is initialized from the policy network $\pi_0$. The formulation follows Generalized Advantage Estimation (GAE) (Schulman et al., 2015).

$$\delta_t = R(s_t, a_t) + V_\phi(s_{t+1}) - V_\phi(s_t),$$

$$\hat{A}_t^\theta(s_t, a_t) = \sum_{t'=0}^{\infty} \lambda^{t'} \delta_{t+t'}, \quad (3)$$

where $\lambda$ is the bias-variance trade-off parameter.

The reward function $R$ reflects the quality of the generated queries, which needs to be consistent with the final evaluation of the task. $\hat{\tilde{x}}$ is fed to the retriever and the reader for a final prediction $\hat{y}$. A part of the reward function is the measures of $\hat{y}$ compared to the golden label $y$ (e.g., exact match and $F_1$ of the predicted answers), denoted as $R_{lm}$. Besides, a KL-divergence regularization is added to prevent the model from deviating too far from the initialization (Ramamurthy et al., 2022; Ziegler et al., 2019).

$$R(s_t, a_t) = R_{lm}(\hat{\tilde{x}}, y) - \beta \mathrm{KL}\,(\pi_\theta \| \pi_0). \quad (4)$$

The final loss function is composed of policy loss and value loss.

$$\mathcal{L}_\theta = -\frac{1}{|\mathcal{S}|\,T} \sum_{\tau \in \mathcal{S}} \sum_{t=0}^{T} \min(k_{t,\theta} A^{\theta'}, \mathrm{clip}\, A^{\theta'}),$$

$$\mathcal{L}_\phi = \frac{1}{|\mathcal{S}|\,T} \sum_{\tau \in \mathcal{S}} \sum_{t=0}^{T} (V_\phi(s_t) - R_t)^2,$$

$$\mathcal{L}_{ppo} = \mathcal{L}_\theta + \lambda_v \mathcal{L}_\phi. \quad (5)$$

Here, $\mathcal{S}$ denotes the sampled set, and $T$ is for step numbers.

## 4  Implementation

**Rewriter**    For the frozen pipeline in §3.1, we prompt an LLM to rewrite the query with few-shot in-context learning (Brown et al., 2020; Min et al., 2022). Our prompt follows the formulation of *[instruction, demonstrations, input]*, where the input is $x$. The instruction is straightforward and demonstrations are 1-3 random examples from training sets and are kept constant across all runs, mainly for the task-specific output format illustration, i.e., a short phrase as an answer for HotpotQA, and an option as an answer for MMLU. For the training scheme in §3.2, we fine-tuning a T5 as the rewriter.

**Retriever**    We use the Bing search engine as the retriever. It requires no candidate index construction like a dense retriever, nor candidates like a textbook. But it allows for a wide knowledge scope and up-to-time factuality. With Bing API, the retrieval is performed in two approaches. (i) For all retrieved web pages, we concatenate the snippets that are related sentences selected by Bing. This method is similar to using a search engine in a browser, input a query and press Enter, then collect the texts shown on the search result page. (ii) For retrieved web pages, we request the URLs and parser to get all the texts. This is similar to clicking on items on the search result page. Then we use BM25 to keep those with higher relevance scores with the query, reducing the document length.

**Reader**    The reader is a frozen LLM, where we adopt ChatGPT (gpt-3.5-turbo) and Vicuna-13B. It performs reading comprehension and prediction with few-shot in-context learning. In our prompt, following the brief instruction and the demonstrations, the input is $x$ or $[doc, \hat{\tilde{x}}]$ with retrieval augmentation.

It has been proved that both the phrasing of prompt lines (Zhang et al., 2023a) and the selection of demonstrations show effects on the in-context learning performance (Su et al., 2022; Zhang et al., 2023b). As it is not the focus of this work, we pay no more attention to prompt editing.

## 5  Experiments

### 5.1  Task Settings

#### 5.1.1  Open-domain QA

Three open-domain QA datasets are used for evaluation. (i) HotPotQA (Yang et al., 2018) consists of complex questions that require multi-hop reasoning. We evaluate the full test set. (ii) AmbigNQ (Min et al., 2020) provides a disambiguated version of Natural Questions (NQ) (Kwiatkowski et al., 2019). For ambiguous questions in NQ, minimal constraints are added to break it into several similar

| **Direct prompt** |
| --- |
| Answer the question in the following format, end the answer with '\*\*'. {demonstration} Question: {$x$} Answer: |
| **Reader prompt in retrieval-augment pipelines** |
| Answer the question in the following format, end the answer with '\*\*'. {demonstration} Question: {$doc$} {$x$} Answer: |
| **Prompts for LLM as a frozen rewriter** |
| *Open-domain QA:* Think step by step to answer this question, and provide search engine queries for knowledge that you need. Split the queries with ';' and end the queries with '\*\*'. {demonstration} Question: {$x$} Answer: *Multiple choice QA:* Provide a better search query for web search engine to answer the given question, end the queries with '\*\*'. {demonstration} Question: {$x$} Answer: |

Table 1: Prompt lines used for the LLMs.

but specific questions. The first 1000 samples are evaluated in the test set. (iii) PopQA (Mallen et al., 2022) includes long-tail distributions as it contains more low-popularity knowledge than other popular QA tasks. We split the dataset into 13k for training and 714 for testing.

Open-domain QA benchmarks are sets of question-answer pairs denoted as $\{(q, a)_i\}$. We use ChatGPT for both the reader and the frozen rewriter. The evaluation metrics are Exact Match ($EM$) and $F_1$ scores. For the reward function in RL, we use an indicator to reward if the retrieved content hits the answer and penalize if misses the answer, denoted as $Hit$. The total reward is a weighted sum of EM, $F_1$, and $Hit$.

$$Hit = \begin{cases} 1 & a \ \text{in} \ doc, \\ -1 & else \end{cases} \quad (6)$$

$$R_{lm} = EM + \lambda_f F_1 + \lambda_h Hit.$$

### 5.1.2 Multiple-choice QA

For multiple-choice QA, our evaluation is conducted on Massive Multi-task Language Understanding (MMLU) (Hendrycks et al., 2021), an exam question dataset including 4 categories: Humanities, STEM, Social Sciences, and Other. Each category is split into 80% for the training set and 20% for the test set.

Multiple-choice QA can be formulated as $\{(q', a)_i\}$, where $q' = [q, c_0, c_1, c_2, c_3]$. $c$ denotes the options, generally there are four for each question. The retrieved documents that are included in the officially provided contaminated lists are ignored. The questions with options are rewritten into search queries. The answer is one option. $EM$ is reported as metrics and used for the reward.

$$R_{lm} = EM. \quad (7)$$

We use ChatGPT as a frozen rewriter and the reader.

We also use Vicuna-13B as the reader for evaluation due to the rate limit issue of ChatGPT. More information on datasets and training setup are presented in the appendix.

### 5.2 Baselines

The following settings are implemented to evaluate and support our methods. (i) **Direct**: The standard in-context learning without any augmentations. (ii) **Retrieve-then-read**: The standard retrieval-augmented method. Retrieved documents are concatenated with the question. (iii) **LLM as a frozen rewriter**: As is introduced in §3.1, we prompt a frozen LLM to reason and generate queries by few-shot in-context learning. (iv) **Trainable rewriter**: Applying the fine-tuned rewriter, the output queries are used by the retriever and the reader. Table 1 presents prompt line forms. Please note that the prompts for prediction are kept the same for each task.

### 5.3 Results

Experimental results on open-domain QA are reported in Table 2. For the three datasets, query rewriting consistently brings performance gain with both a frozen rewriter and a trainable rewriter. On AmbigNQ and PopQA, the standard retrieval augments the reader, indicating useful external knowledge is retrieved. On HotpotQA, the standard retrieval hurts the reader. This shows that using complex questions as queries cannot compensate for the parametric knowledge, but bring noises instead (Mallen et al., 2022). This suggests that multi-hop questions are not suitable queries for the web search engine. The scores increase by adding the rewriting step. On PopQA, our trainable rewriter surpasses standard retrieval while being inferior to the LLM rewriter. This indicates that the

distillation of query rewriting is sub-optimal.

The scores on multiple-choice QA are presented in Table 3. With ChatGPT as a reader, it can be observed that query rewriting improves the scores in most of the settings, except for the social sciences category. With Vicuna as a reader, our method achieves more gains on the four categories compared to ChatGPT. This agrees with the intuition that a more powerful reader has more parametric memories, thus more difficult to compensate with external knowledge.

| Model | EM | $F_1$ |
|---|---|---|
| *HotpotQA* | | |
| Direct | 32.36 | 43.05 |
| Retrieve-then-read | 30.47 | 41.34 |
| LLM rewriter | 32.80 | 43.85 |
| Trainable rewriter | 34.38 | 45.97 |
| *AmbigNQ* | | |
| Direct | 42.10 | 53.05 |
| Retrieve-then-read | 45.80 | 58.50 |
| LLM rewriter | 46.40 | 58.74 |
| Trainable rewriter | 47.80 | 60.71 |
| *PopQA* | | |
| Direct | 41.94 | 44.61 |
| Retrieve-then-read | 43.20 | 47.53 |
| LLM rewriter | 46.00 | 49.74 |
| Trainable rewriter | 45.72 | 49.51 |

Table 2: Metrics of open-domain QA.

| MMLU | EM | | | |
|---|---|---|---|---|
| | Human. | STEM | Other | Social |
| *ChatGPT* | | | | |
| Direct | 75.6 | 58.8 | 69.0 | 71.6 |
| Retrieve-then-read | 76.7 | 63.3 | 70.0 | 78.2 |
| LLM rewriter | 77.0 | 63.5 | 72.6 | 76.4 |
| *Vicuna-13B* | | | | |
| Direct | 39.8 | 34.9 | 50.2 | 46.6 |
| Retrieve-then-read | 40.2 | 39.8 | 55.2 | 50.6 |
| LLM rewriter | 42.0 | 41.5 | 57.1 | 52.2 |
| Trainable rewriter | 43.2 | 40.9 | 59.3 | 51.2 |

Table 3: Metrics of multiple choice QA.

# 6 Analysis

## 6.1 Training Process

The training process includes two stages, warm-up and reinforcement learning. This section shows the validation scores of the three open-domain QA datasets for further analysis. Figure 2 presents the metric scores through training iterations in the process of reinforcement learning. As the rewriting models have been warmed up on the pseudo data before RL, scores at "0 iteration" denote the ability acquired from the warm-up training.

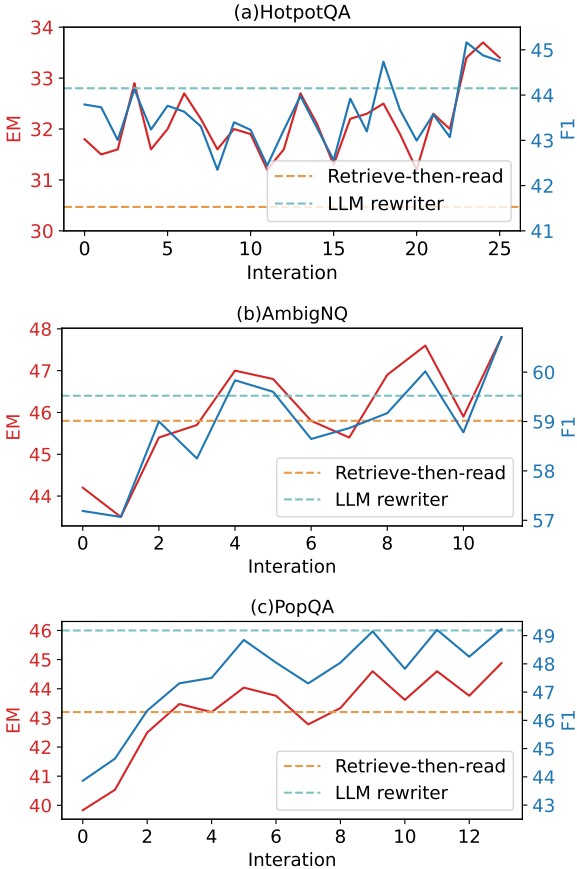

Figure 2: Reinforcement learning validation scores of (a)HotpotQA, (b)AmbigNQ, and (c)PopQA. The solid lines show EM (red) and F1 (blue) numbers through training iterations. The dashed lines are EM scores of the standard retrieve-then-read method (orange) and retrieval with an LLM as the rewriter (green).

It can be observed that the curves show upward trends with some fluctuations on all the datasets. (i) For multi-hop questions in HotpotQA, the standard retrieval is relatively weaker. Complex questions can be not specific search queries and show a larger gap from rewritten queries, i.e., the green and red lines. (ii) On AmbigNQ and PopQA, our method surpasses the baselines after several iterations (3 or 4). This indicates that the RL training stage can compensate for the insufficiency of the distillation on the pseudo data during warm-up training. (iii) In particular, on PopQA, the trainable rewriter remains inferior to the LLM rewriter. This can be explained as the dataset is constructed for adaptive retrieval (Mallen et al., 2022), which only uses retrieval where it helps to avoid harmful redundant retrieval. Thus, *"None"* is a possible query that means no retrieval. This causes more complexity and uncertainty. LLM rewriter knows better when the retrieval is needed for itself as a reader, although the rewriting step is not concatenated as

the input context of the reader.

We calculate the performance of query *"None"*. The questions that can be correctly answered without retrieval (i.e., the "Direct" method) are those samples that need no more context. Comparing this retrieval-free set with those that are rewritten to be *"None"* query, the $F_1$ score of the LLM rewriter is 71.9% and the T5 rewriter score is 67.1%. If we consider the questions that can be correctly answered without retrieval but go wrong with retrieval as the retrieval-free set, the $F_1$ scores are 78.7% for LLM rewriter and 77.4% for T5.

| Model | EM | $F_1$ | Hit ratio |
|---|---|---|---|
| No retrieval | 42.10 | 53.05 | – |
| Upper bound | 58.40 | 69.45 | 100 |
| *Retrieve-then-read* | | | |
| w/ snippet | 38.70 | 50.50 | 61.1 |
| w/ BM25 | 45.80 | 58.50 | 76.4 |
| *LLM rewriter* | | | |
| w/ snippet | 39.80 | 52.64 | 63.5 |
| w/ BM25 | 46.40 | 58.74 | 77.5 |
| *Trainable rewriter* | | | |
| w/ BM25[2] | 47.80 | 60.71 | 82.2 |

Table 4: Retrieval analysis on AmbigNQ.

## 6.2 Retrieval Result

Our proposed method is a pipeline framework, instead of an end-to-end system. The query rewriting first affects the retrieved context, then the context makes a difference to the output of the reader. Hence, QA metrics are indirect measurements. We take a closer look at the retrieved context and the reader capability through the retrieval metric, hit ratio. After text normalization, the hit rate is computed to measure whether the retrieved context contains the correct answers.

Table 4 shows the scores on AmbigNQ. The scores in the second line are computed on a selection of the samples whose retrieved contexts hit correct answers (under the standard retrieve-then-read setting). The scores show the approximate upper bound ability of the reader with retrieval augmentation, abbreviated as the "upper bound" score. The effectiveness of retrieval is proved compared to the no retrieval setting (the first line). For each retrieval method, two settings are presented: (i) collecting Bing snippets, (ii) selecting from URLs by BM25. The metrics show that content selection with BM25 recalls better documents than snippets,

---

[2]Our trainable rewriter is adapted to the retriever using BM25 during RL training. Using the output queries of the test set after training, the snippet hit rate is 73.4%.

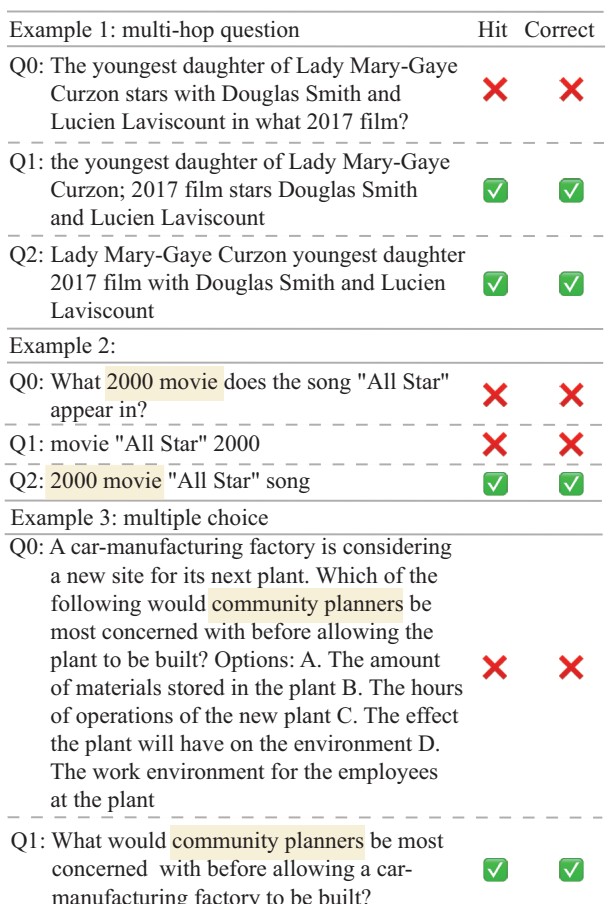

Figure 3: Examples for intuitive illustration. Q0 denotes original input, Q1 is from the LLM rewriter, and Q2 is from the trained T5 rewriter. **Hit** means retriever recall the answer, while **Correct** is for the reader output.

while query rewriting makes progress on both settings. We also observed that the improvement in the hit rate of the retriever is more significant than the improvement in the reader. This is consistent with the findings in related search (Mallen et al., 2022; Liu et al., 2023).

## 6.3 Case Study

To intuitively show how the query rewriting makes a difference in the retrieved contexts and prediction performance, we present examples in Figure 3 to compare the original questions and the queries. In example 1, the original question asks for a film that *the youngest daughter of Lady Mary-Gaye Curzon* co-stars with two certain actors. Both query 1 and query 2 put the keyword *film* forward, closely following *the youngest daughter of Lady Mary-Gaye Curzon*. With both, the actress *Charlotte Calthorpe* and her movie information can be retrieved and the answer is included. The second is an example where the query from the LLM rewriter failed but

the query from T5 gets the correct answer. The number *2000* is misunderstood in query 1, while query 2 keeps *200 movie* together, avoiding meaningless retrieval. Example 3 is for multiple choice. The query simplifies the background and enhances the keyword *community planner*. The retrieve contexts are mainly about *Introduction to Community Planning* where the answer *environment* appears several times.

## 7   Conclusion

This paper introduces the *Rewrite-Retrieve-Read* pipeline, where a query rewriting step is added for the retrieval-augmented LLM. This approach is applicable for adopting a frozen large language model as the reader and a real-time web search engine as the retriever. Further, we propose to apply a tuneable small language model the rewriter, which can be trained to cater to the frozen retriever and reader. The training implementation consists of two stages, warm-up and reinforcement learning. Evaluation and analyses on open-domain QA and multiple-choice QA show the effectiveness of query rewriting. Our work proposes a novel retrieval-augmented black-box LLM framework, proves that the retrieval augmentation can be enhanced from the aspect of query rewriting, and provides a new method for integrating trainable modules into black-box LLMs.

## Limitations

We acknowledge the limitations of this work. (i) There is still a trade-off between generalization and specialization among downstream tasks. Adding a training process, the scalability to direct transfer is compromised, compared to few-shot in-context learning. (ii) The research line of *LLM agent* has shown impressive performance but relies on multiple calls to the LLM for each sample (Khattab et al., 2022; Yao et al., 2023), where the LLM plays as an agent to flexibly call the retriever multiple times, reads the context in earlier hops, and generates follow-up questions. Different from these studies, our motivation is to enhance the one-turn retriever-then-read framework with a trainable query rewriter. (iii) Using a web search engine as the retriever also leads to some limitations. Neural dense retrievers that are based on professional, filtered knowledge bases may potentially achieve better and controllable retrieval. More discussion is included in the appendix.

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

## A  Warm-up Dataset

For the warm-up training of the tuneable rewriter, we construct a pseudo dataset for the query rewriting task. For benchmarks that provide official training and test splits (HotpotQA and AmbigNQ), we use the whole training set. For those that have no official splits (PopQA and MMLU), we randomly split the full dataset. In detail, PopQA contains 16 types of questions, thus split into 13k for training and 714 for testing following stratified sampling. For MMLU, each of the 4 categories is randomly split into 80% for the training set and 20% for the test set. Then the training sets of each benchmark are used to derive the pseudo dataset for the query rewriting, i.e., $D_{Train} = \{(x, \tilde{x}) | \hat{y} = y\}$. We present the statistics of the splits and warm-up dataset in Table 5.

## B  Setup Details

For warm-up, we train the T5-large with 3e-5 learning rate, {16, 20} batch size, for {6,8,12} epochs. For reinforcement learning, we set the sampling

| Task | Training Set | Warm-up | Test Set |
|------|-------------|---------|----------|
| HotpotQA | 90.4k | 37.5k | 7.4k |
| AmbigNQ | 19.4k | 8.6k | 1k |
| PopQA | 13.0k | 6.0k | 0.7k |
| Humanities | 3.8k | 1.5k | 0.9k |
| STEM | 2.4k | 0.9k | 0.6k |
| Other | 2.6k | 1.3k | 0.6k |
| Social Science | 2.4k | 1.3k | 0.6k |

Table 5: Metrics of multiple choice QA.

steps to 5120, 10 threads, 512 steps for each. After sampling, the policy network is trained for {2,3,4} epochs, with learning rate as 2e-6 and batch size as {8,16}. $\lambda_f$ and $\lambda_h$ are 1.0. $\beta$ in Eq. 4 is dynamically adapted according to Ramamurthy et al. (2022); Ziegler et al. (2019),

$$e_t = \text{clip}\left(\frac{\text{KL}(\pi \| \pi_0) - \text{KL}_{\text{target}}}{\text{KL}_{\text{target}}}, -0.2, 0.2\right),$$
$$\beta_{t+1} = \beta_t (1 + \text{K}_\beta e_t),$$

where $\text{KL}_{target}$ is set to 0.2, $\text{K}_\beta$ is set to 0.1. $\beta_0$ is initialized to be 0.001. The generation strategy follows the 4-beam search and returns the one sequence. In the implementation of the BM25-based retriever, the textboxes from searched URLs are parsed from HTML code. We compute BM25 scores between the paragraph from each textbox and the query following the scikit-learn package, then keep those with higher scores until the reserved context reaches a max length. In reinforcement learning, the results of AmbigNQ are with the BM25 method, while others use snippets as context.

## C  Web Search: Tool Use

Our proposed pipeline integrates an externally built web search engine as the retriever module. We present more discussion on the advantages and disadvantages here.

The usage of external tools expands the ability boundary of language models, compensating for the parametric knowledge, and grounding the capabilities of language models to interact with environments (Qin et al., 2023; Schick et al., 2023). Recent studies show a trend to leverage plug-and-play tools like search engines to enhance language agents (Lazaridou et al., 2022; Menick et al., 2022; Shuster et al., 2022; Shen et al., 2023). Search engine APIs are well-developed retrievers, saving efforts to build and maintain another retriever, like a Contriever. Accessible to the whole Internet, the web search retrieves from a wide-range, up-to-date

knowledge base. The temporal misalignment problem on a fixed candidate database can be alleviated.

On the other hand, web search APIs are commercial products requiring subscriptions. Also, the vast amount of knowledge on the web can be difficult to control. The retrieved context from the Internet can be occasionally inconsistent, redundant, and toxic, which hinders the LLM reader.

Beyond retrieval augmentation, in a general scope, other tools called by LLMs, like code interpreters, online models, and expert applications, are all similar to search engines, without trainable parameters to optimize. There could be a gap between the LM and these tools. This paper proposes an idea to align them through a trainable small model.