# OpenReview forum: "Query Rewriting in Retrieval-Augmented Large Language Models"
_EMNLP/2023/Conference — EMNLP 2023 Main_

### Official Review · Reviewer_JBWk · 2023-08-02

**Soundness:** 2

**Excitement:**

2: Mediocre: This paper makes marginal contributions (vs non-contemporaneous work), so I would rather not see it in the conference.

**Missing References:**

Please see above.

**Paper Topic And Main Contributions:**

This paper proposes a Rewrite-Retrieve-Read pipeline for in-context learning. The essence of the idea is to allow an LM to rewrite incoming queries to make them more amenable to effective retrieval-augmented generation. The paper explores two rewriting approaches: one using a frozen LLM and one using a custom T5-large module trained with reinforcement learning. Overall, the approach yields solid results, with the best results coming from the approach that trains a custom query rewriter.


**Questions For The Authors:**

What is the relative contribution of the warm-up and RL phases for the trained rewriter approach?


**Reasons To Accept:**

This is a creative paper with informative evaluations. I really like the idea of having small, custom built models for in-context learning systems, as an alternative to relying on LLMs for everything.


**Reasons To Reject:**

1. The paper says, "rewrite-retrieve-read is the first framework where input text is adapted for the frozen retriever and LLM reader." This is not correct. The DSP paper, which the authors cite, uses exactly this sort of approach, and for very similar tasks. DSP also does query rewriting for conversational QA. Self-Ask is another approach where the query is iteratively rewritten to improve retrieval and make the job easier for the LLM.

2. The DSP paper also evaluates on HotPotQA. The DSP program does iterative query rewriting to break the question down. The EM reported is 51.4% EM, which is substantially higher than the best result reported in this paper: 34.38% EM. The DSP baseline retrieve-then-read system is at 36.9. I am fine with the idea that the goal of the current paper is to test query rewriting, and so maybe we don't expect SOTA overall, but the DSP approach is so similar that I can't see a justification for excuding it, and the fact that the retrieve-then-read baseline is stronger is certainly informative.

3. It is not surprising that a trainable rewriter does better than a system that has to reply on frozen components. That said, the DSP program that got 51.4% EM on HotPotQA uses only frozen components.

4. The retrieval engine for the paper is a Web search engine. I suppose this has independent interest, but it is a challenge for reproducibility, and the authors themselves seem to indicate that it was a suboptimal choice (line 477). It might have been more productive to use a retriever and retrieval model for the relevant domains.

**Reproducibility:**

2: Would be hard pressed to reproduce the results. The contribution depends on data that are simply not available outside the author's institution or consortium; not enough details are provided.

**Reviewer Confidence:**

4: Quite sure. I tried to check the important points carefully. It's unlikely, though conceivable, that I missed something that should affect my ratings.

---

> ### Author Rebuttal · Authors · 2023-08-29
>
> Thank you for your time and careful review.
> (Notations: R: Reason to Reject; Q: Question for the authors)
>
> **R1: DSP & Self-ask**
>
> DSP and Self-ask are excellent works on retrieval-augmented LLM. But we clarify that our method differs from these works.
>
> 1) Our main idea is the trainable pipeline, adding a trainable small model rather than relying on the expensive, frozen LLM for everything.
> 2) The rewriting in their methods relies on multiple interactions with the LLM (#Line 76-82), e.g., to read the context retrieved in earlier hops and generate follow-up questions, which requires multiple calls to the LLM for each sample. Their approaches are closer to the *LLM agent* research line (like ReAct, etc.), from which our motivation differs. Our motivation is to enhance the retriever-then-read framework with a trainable query rewriter.
>
> Our statement (#Line 133-135) does not intentionally ignore contributions of previous work. What we tried to emphasize is that our proposed framework is the first to adapt the input text by a trainable rewriter to the frozen retriever and LLM reader (Similar expressions are in #Line 271-273, 114-117).
>
> We will improve the expression to avoid misunderstanding.
>
>
> **R2&R3: DSP scores**
>
> DSP achieves impressive scores on HotpotQA, which is in no contradiction to that our experiments can demonstrate the effectiveness of our proposed method.
>
> 1) As mentioned in our reply to R1, DSP emphasizes sophisticated multiple interactions with the LLM agent. For each question, the LLM is called iteratively for each action step and takes an action based on previous retrieval results. In contrast, only focusing on query rewriting, our method only calls LLM once as the reader (and extra once if the LLM is used as the rewriter).
>
> 2) DSP also includes a demonstration enhancement step with a 16-shot training set, while we use unchanged random 1-3 demos (#Line 371-372).
>
> 3) ChatGPT API being a limited resource, related works often use different evaluation settings for different approaches. DSP is tested on 1000 samples while we use the total 7405 test samples.
>
> We hope our above responses could clarify our novel motivation and the differences from related works.
>
> **R4: Web search**
>
> In #Line 477, we want to express that using original questions in HotpotQA as the retrieval query is not suitable, and a rewriting step is required, as mentioned in motivation (#Line 86-92).
>
> Web search does have pros and cons (#Line 600) but has been a widely used retrieval tool for LLMs (#Line 174-187). The proposed method is applicable to all sparse retrievers, while web search has advantages as a light sparse retriever with a wide knowledge source.
> We thank you for this question and will add more discussion to our paper.
>
> **Q1: Warm-up and RL phases**
>
> This is discussed in Section 6.1. Figure 2 shows the scores during RL, where the start points (0 iteration) show the scores after warm-up. The warm up step is for a warm start, where the rewriter is inferior to the LLM rewriter. Although the distillation on the pseudo data is insufficient, the RL phase compensates for the gap.
>
> **Missing References**
>
> Thank you. We will add the paper of Self-ask as a reference.

---

### Official Review · Reviewer_rmM6 · 2023-08-09

**Soundness:** 3

**Excitement:**

3: Ambivalent: It has merits (e.g., it reports state-of-the-art results, the idea is nice), but there are key weaknesses (e.g., it describes incremental work), and it can significantly benefit from another round of revision. However, I won't object to accepting it if my co-reviewers champion it.

**Paper Topic And Main Contributions:**

This paper adapts the search query to improve the retrieval augmented system by adding a query rewriting step and freeze the LM and the retriever. The methods trains a smaller model as the rewriter. The rewriter is first warmed up by training on pseudo rewrites that leads to  correct y, and then finetuned via reinforcement learning, with the reward function be a task specific metric (like Exact Match). The methods is evaluated on open-domain QA, and multiple choice QA, and show empirical gains.

**Reasons To Accept:**

1. The problem to tackle is well-motivated.
2. The proposed approach is solid and reasonable.
3. the experiments are quite convincing with the case study + analysis are quite convincing.


**Reasons To Reject:**

1. I'm concerned about the novelty of the method, and how this way of learning to rewrite the query compares to other approaches that finetunes the reader or the retriever, in compute/performance tradeoffs.

**Reproducibility:**

4: Could mostly reproduce the results, but there may be some variation because of sample variance or minor variations in their interpretation of the protocol or method.

**Reviewer Confidence:**

3: Pretty sure, but there's a chance I missed something. Although I have a good feel for this area in general, I did not carefully check the paper's details, e.g., the math, experimental design, or novelty.

---

> ### Author Rebuttal · Authors · 2023-08-29
>
> Thank you for your careful review and valuable feedback.
> (Notations: R: Reason to Reject; Q: Question for the authors)
>
> **R1: Novelty & compute/performance tradeoffs**
>
> We clarify the novelty and compute/performance tradeoffs compared with REPLUG, the first work to adapt a dense retrieval model to the LM.
>
> **Novelty:**
>
> Our work enhances the retrieval-augmented LLM from totally another perspective. REPLUG finetunes a dense retriever, while our method finetunes a rewriter to align the query to the retriever and reader.
>
> Compared to REPLUG, our novel motivation is in two folds: (1) Input texts themselves need to be optimized (#Line 86-92). (2) Query rewriting can align queries to sparse, frozen retrievers, avoiding the maintenance of the search index over time (#Line 99-102).
>
> **Compute/performance tradeoffs:**
>
> In the training process, REPLUG finetunes a Contriever (BERT-base) with a 36M-candidate set and performs an ensemble between top-20 docs. Our method finetunes a rewriter (T5-large), taking about 0.5-1.5 hours for each epoch (including many retries due to the rate limit issue of ChatGPT, which causes extra delay).
>
> In terms of performance, compared to the approach with frozen pipeline, the trainable Contriever improves open-domain QA by 1.22% on average and MMLU by 0.56%. Our trainable T5 rewriter improves open-domain QA by 2.41% on average and MMLU by 0.84%.

---

### Official Review · Reviewer_GP32 · 2023-08-11

**Soundness:** 4

**Excitement:**

3: Ambivalent: It has merits (e.g., it reports state-of-the-art results, the idea is nice), but there are key weaknesses (e.g., it describes incremental work), and it can significantly benefit from another round of revision. However, I won't object to accepting it if my co-reviewers champion it.

**Missing References:**

The authors mention reasoning errors of retriever-based models in Section 3.2, which can benefit from including insights from:

BehnamGhader, Parishad, Santiago Miret, and Siva Reddy. "Can retriever-augmented language models reason? the blame game between the retriever and the language model." arXiv preprint arXiv:2212.09146 (2022).

**Paper Topic And Main Contributions:**

The papers proposes a new method for retriever augmented LLMs that includes query rewriting, which changes the pipeline of retriever-based LLMs from retrieve-the-read to rewrite-retrieve-read. The authors first introduce some of the common problems with LLMs that can be addressed by retriever-based models, such as hallucinations and factuality errors. Next, the authors describe some shortcomings with the currently applied retrieve-then-read pipeline and propose their rewrite-retrieve-read pipeline that involves query rewriting by an additional LLM in the loop of retriever-based models. Next, the authors describe related work on retriever LLMs, using internet search queries as a knowledge base and interactions with black-box commercial LLMs with APIs.

In the methods section, the paper describes the important components of retriever based language models, as well as their method. This includes the training scheme for query rewriting with a warm-up objective and a reinforcement learning (PPO) based training. The authors also describe implementation details of their method, including model types used for each part of the pipeline, as well as different training settings (e.g. trainable and non-trainable query rewriters).

In their experiments, the authors test their method on various QA datasets, including HotPotQA, AmbigQA and PopQA for open-domain QA and MMLU for multiple-choice QA, all of which test different capabilities of the pipeline. In their results, the authors show that rewriter queries for retriever-based models generally improves performance for open-domain QA and for multiple choice QA (with the exception of social science). The authors of their results, including training details, retrieval abilities and a case study. The analysis is followed by a conclusions and a brief discussion of limitations.

**Questions For The Authors:**

* Question A: Can you describe how you obtain the demonstrations for your prompts that you mention in Table 1 and various parts of the paper?
* Question B: Can you provide a description of BM25? It is currently missing from the paper based on what I can tell and would be good to clarify.
* Question C: Does your RL training procedure apply a dense or sparse reward function?
* Question D: Can you explain the fluctuations in performance in Figure 2? It appears that performance varies a lot and it's a bit hard to tell why that might be happening.

**Reasons To Accept:**

* The paper proposes a new method for retriever-based LLMs changing the pipeline from retrieve-then-read to rewrite-retrieve-read. The method, and relevant details, are generally well described and clearly laid out.
* The results presented in the experiments generally support the notion that query rewriting improves performance of LLMs on a diverse set of QA benchmarks testing.
* The authors provide an analysis of that provide further insight into the inner working of the proposed method, including specific query rewriting examples and a discussion when training a rewriter may or may not be beneficial.
* The paper is generally well written with relevant descriptions of methods, experiments, results and analysis.

**Reasons To Reject:**

* It is unclear if retrieval through a web API can provide consistent results given that the results of API queries could change. It would be good for the authors to include a discussion on this.
* The motivation for choosing different architectures as different parts of the pipeline could be more detailed, as could the details of the PPO-based training procedure (e.g. how many samples were used for training and refinement and how was convergence determined).
* The query rewriting examples in Figure 3 show only minimal changes to the text of the query. This makes the effect of the rewriter a bit nebulous. A way of showing the impact of the rewriter on changing the queries (other than final results) could be useful.

**Reproducibility:**

4: Could mostly reproduce the results, but there may be some variation because of sample variance or minor variations in their interpretation of the protocol or method.

**Reviewer Confidence:**

4: Quite sure. I tried to check the important points carefully. It's unlikely, though conceivable, that I missed something that should affect my ratings.

**Typos Grammar Style And Presentation Improvements:**

* Line 099: We adopt *an*
* Line 188: Cooperate with Black-box LLMs -> Cooperation with Black-box LLMs
* Line 398: proved -> proven

---

> ### Author Rebuttal · Authors · 2023-08-29
>
> Thank you for your careful review and valuable feedback.
> (Notations: R: Reason to Reject; Q: Question for the authors)
>
> **R1: Search results**
>
> We really appreciate this advice. Web search retrieves flexible, various external knowledge (#Line 600). In our preliminary experiments, the Hit scores are relatively consistent.
>
> As a convenient tool, web search API has been widely adopted as a plug-and-play, auxiliary component for LM-based systems (#Line 174-187). Following these methods, we apply web search as a light sparse retriever, while our proposed method is applicable to all sparse retrievers. We will surely expand experiments and present more detailed analyses.
>
> **R2: Details about choosing different architectures & PPO-based training procedure**
>
> **Architectures:**
> Relevant content is scattered in the paper.
>
> Reader: Our main idea lies in tunable augmentation for LLMs. ChatGPT is an appropriate foundation model. Vicuna-13B is an acceptable alternative that our computing resources can afford (#Line 450).
>
> Retriever: Web search is applied as it is a flexible and light retriever, and the Internet is a wide, open knowledge source avoiding search index reconstruction (#Line 376, 174-187).
>
> Rewriter: (1) Using ChatGPT as a rewriter provides an unsupervised upper bound, and is a preliminary verification for the proposed pipeline. (2) For the trainable rewriter, T5 is a seq2seq LM pre-trained with a task-specific prefix. We define the prefix “rewrite a better search query” to tune T5 to our rewriting task.
>
> **PPO:**
> The concerned training settings are addressed as follows and will be strengthened in the later version of this paper.
>
> HotpotQA and AmbigQA provide both training and test splits, we use the whole training set. For PopQA and MMLU, we randomly split them for training and evaluation (#Line 422, 438). We present the statistics of the training splits and warmup data (#Line 293-295) here.  The convergence is observed by comparing the changes in training loss and validation score.
>
> We will add details to the paper and release the datasets.
>
> |	   |HotpotQA	|AmbigQA	|PopQA	|Human	|STEM	|Other	|Social|
> |-------|---------------|----------------|-----------|-----------|-----------|-----------|-------|
> |Training	|90.4k	|19.4k	        |13k 	|3.8k	|2.4k	|2.6k	|2.4k|
> |Warmup	|37.5k	|8.6k	        |6k	        |1.5k	|0.9k	|1.3k	|1.3k|
>
> **R3: Query rewriting examples**
>
> In Figure 3, the rewritten queries show differences compared to the original inputs (Q0), The examples mainly show the changes of redundancy removing, swapping, and keyword reserving.
> We thank you for this advice and will improve the case study.
>
> **QA: Demonstrations**
>
> The demonstrations are random samples from training set and are kept constant across all runs. The demonstrations are mainly for output format illustration (#Line 371-372), i.e., a short phrase as an answer for HotpotQA, and an option as an answer for MMLU.
>
> **QB: BM25**
>
> Thank you. We will add more detailed descriptions.
>
> For each searched URL, the textboxes are parsed from HTML code. We compute BM25 scores between the paragraph from each textbox and the query (following scikit-learn package), then keep those with higher scores until the reserved context reaches a max length (#Line 386-390).
>
> **QC: Dense or sparse reward function**
>
> This is a good question as the sparse reward problem indeed exists, however, it does not matter too much for our proposed solution due to the following detailed settings. 1) The output of query rewriting is not long. There is relatively small room in need for exploration.
> 2) The warmed-up rewriter has acquired some domain knowledge.
>
> **QD: Fluctuations in Figure 2**
>
> The fluctuations mostly occur in HotpotQA, which can be attributed to the uncertain hop step numbers that bring about extra reasoning difficulty.
>
> **Missing References & Typos Grammar Style And Presentation Improvements**
>
> Thanks very much for your advice. We will add this reference and correct the typos.

---

### Official Review · Reviewer_ksRg · 2023-08-13

**Soundness:** 4

**Excitement:**

3: Ambivalent: It has merits (e.g., it reports state-of-the-art results, the idea is nice), but there are key weaknesses (e.g., it describes incremental work), and it can significantly benefit from another round of revision. However, I won't object to accepting it if my co-reviewers champion it.

**Paper Topic And Main Contributions:**

The paper focuses on how to best optimize the retrieved passages to augment the LLM and obtain best QA performance.

The main contribution of this paper is an NLP engineering experiment.

The paper shows improvements in QA performance with the proposed rewrite-retrieve-read QA approach. The approach entails rewriting the query to retrieve documents that better fit the question. Two rewriters are used - a frozen LLM rewriter and a smaller T5 model trained using RL to perform query rewriting based on rewards computed using LLM reader feedback.

**Questions For The Authors:**

Question A: What is the retriever HIT rate when using the Trainable T5 query rewriter?
Question B: What are the specifics of the warmup dataset, number of samples, etc?

**Reasons To Accept:**

The paper proposed a new cascaded system that rewrites queries to retrieve better documents for improved performance on QA tasks using an LLM reader.

The smaller T5 rewriter model warmed up on a specially curated pseudo data and fine-tuned using LLM feedback as the RL reward outperformed the more compute intensive LLM query rewriter in all QA tasks except PopQA.



**Reasons To Reject:**

The paper's results is currently not reproducible. No information has been provided on the curated dataset for the trainable T5 warmup.

No mention of the in context examples used to prompt the model to rewrite the query. Only mentioned that random samples are used, but no information on whether the demonstrations are samples randomly for each run or kept constant across all runs.

Need more details for reproducibility of the results.

Results for retriever HIT rate when using the Trainable T5 query rewriter not provided.

**Reproducibility:**

2: Would be hard pressed to reproduce the results. The contribution depends on data that are simply not available outside the author's institution or consortium; not enough details are provided.

**Reviewer Confidence:**

3: Pretty sure, but there's a chance I missed something. Although I have a good feel for this area in general, I did not carefully check the paper's details, e.g., the math, experimental design, or novelty.

**Typos Grammar Style And Presentation Improvements:**

Overall peer easy to understand. Few grammatical mistakes, would suggest another review.

However, Analysis section was a bit difficult to understand and required multiple reading passes to get clarity on the point trying to be conveyed, especially section 6.1

---

> ### Author Rebuttal · Authors · 2023-08-29
>
> Thank you for your careful review and valuable feedback.
> (Notations: R: Reason to Reject; Q: Question for the authors)
>
> **R1&QB: Dataset for the trainable T5 warmup**
>
> The dataset is introduced in sections 3.2.1 and 5.1.
>
> For benchmarks that provide official training and test splits (HotpotQA and AmbigQA), we use the whole training set; For others (PopQA and MMLU), we randomly split them (#Line 422, 438).
>
> For warmup, the training set is used to derive pseudo dataset as described in #Line 288-295.
>
> We present the statistics of the training splits and warmup data here. We will add this to the paper and release the datasets.
> |	   |HotpotQA	|AmbigQA	|PopQA	|Human	|STEM	|Other	|Social|
> |-------|---------------|----------------|-----------|-----------|-----------|-----------|-------|
> |Training	|90.4k	|19.4k	        |13k 	|3.8k	|2.4k	|2.6k	|2.4k|
> |Warmup	|37.5k	|8.6k	        |6k	        |1.5k	|0.9k	|1.3k	|1.3k|
>
> **R2: Random demonstrations**
>
> The demonstrations are from training sets and are kept constant across all runs. The demonstrations are mainly for output format illustration (#Line 371-372), i.e., a short phrase as an answer for HotpotQA, and an option as an answer for MMLU.
>
> **R3: Details for reproducibility**
>
> Thanks for your advice. Some experiment setups are presented in Appendix A. We surely will add more details to the paper and release our code, results, and data.
>
> **R4&QA: HIT rate with trainable T5 query rewriter**
>
> The Hit score is originally computed for preliminary experiments to prove the effectiveness of query rewriting and to provide an upper bound. To reply to this question, we conduct a quick experiment on AmbigQA with the T5 rewriter, and the best hit scores are 82.2 with BM25 and 70 with snippet.
>
> We thank you for this question and will add more Hit scores.
>
> **Typos Grammar Style and Presentation Improvements**
>
> Thank you for your careful reading. We will improve the writing.

---

### Meta-Review · Area_Chair_BaPK · 2023-09-15

**Recommendation:** 5

**Metareview:**

This paper proposes a rewrite-retrieve-read approach as opposed to a retrieve-read approach in order to refine the retrieval query to be tailored toward the needed knowledge that should be retrieved from the knowledge source. The rewriter model is learned using reinforcement learning where the reward is based on the end prediction from the language model. Evaluation on three open-domain QA datasets and 1 multi-choice dataset reveals the effectiveness of the method.

The reviewers acknowledged that the contribution is in proposal of a new method that is shown to be effective (ksRg, GP32, rmM6), experiments and analysis are thorough (GP32, rmM6), and the paper is well-written (GP32).

Reviewers raised concerns on heavy reliance on close-sourced models such as web API (GP32, JBWk) and qualitative analysis (rewriting seems minor based on examples) (GP32). Also, reviewers pointed out lack of details reported in the paper (ksRg) and lack of comparison to prior work (rmM6, JBWk). These concerns are sufficiently addressed by authors during the discussion period. Authors are encouraged to add details and discussion in the final version of the paper if accepted.

---

### Decision · Program_Chairs · 2023-10-07

**Decision:**

Accept-Main

**Comment:**

This paper proposes a rewrite-retrieve-read approach as opposed to a retrieve-read approach in order to refine the retrieval query to be tailored toward the needed knowledge that should be retrieved from the knowledge source. The rewriter model is learned using reinforcement learning where the reward is based on the end prediction from the language model. Evaluation on three open-domain QA datasets and 1 multi-choice dataset reveals the effectiveness of the method.

The reviewers acknowledged that the contribution is in proposal of a new method that is shown to be effective (ksRg, GP32, rmM6), experiments and analysis are thorough (GP32, rmM6), and the paper is well-written (GP32).

Reviewers raised concerns on heavy reliance on close-sourced models such as web API (GP32, JBWk) and qualitative analysis (rewriting seems minor based on examples) (GP32). Also, reviewers pointed out lack of details reported in the paper (ksRg) and lack of comparison to prior work (rmM6, JBWk). These concerns are sufficiently addressed by authors during the discussion period. Authors are encouraged to add details and discussion in the final version of the paper if accepted.